# SYSIDBENCH: A BENCHMARK FOR SYSTEM IDENTIFICATION METHODS

## ABSTRACT

Modeling the behaviour of dynamic systems is a difficult problem because (i) there is a plenitude of existing system identification methods and (ii) the broadly varying characteristics of different dynamic systems are not all addressed by a single best method. While benchmarking of system identification methods has been recognized to constitute an important asset for developers who want to select the most suitable method for their problem, these benchmarks currently lack capabilities that developers require for systematic benchmarking. Analysing related work and our own, we have worked out five requirements on the benchmarking of system identification methods that have shaped the design of SYSIDBENCH, our novel benchmark, which comprises data sets with specifically tailored data types, data splits and evaluation metrics. In particular, SYSIDBENCH comprises a principle-based summarizing evaluation metrics using predictions of energy as key measurement target, it allows for judging generalization capabilities of system identification methods, and it investigates the fulfillment of physical principles. The code for our benchmark, including the links to the datasets, is available at `anonymous.github.repository`

## 1 INTRODUCTION

Given an initial state $x^0 \in \mathbb{R}^n$ and an input signal $u := (u^k)_{k=0,...,N-1}$, dynamic system models predict the output signal $\hat{y} := (\hat{y}^k)_{k=0,...,N-1}$ of a physical system. System identification algorithms determine the model parameters such that the predicted output signals closely match the output signals from the physical, but unknown, system, for example, the dynamics of a ship (Baier & Staab, 2022), the vibration of an aircraft wing (Noël & Schoukens, 2020a), or the fluid flow in the wake of a moving cylinder (Decuyper et al., 2024). The parametrized models are used for simulation, controller design, or as a sub-model of a complex system.

When selecting an algorithm for parameterizing such models from input-output sequences, users are facing the challenge of choosing from many different algorithms stemming from the systems and control (Pillonetto et al., 2022; Forgione & Piga, 2021b; Beintema et al., 2023b;a) as well as the machine learning community (Baier et al., 2023; Mohajerin & Waslander, 2019; Hu et al., 2024; Bonassi et al., 2024; Gu et al., 2021). The choice is further complicated by the different objectives the user may impose on the resulting model. When the model is used for simulation, one is interested in highly accurate multi-step predictions for previously unseen input signals. For controller design, one aims for closed-loop guarantees, where the model is in a feedback-interconnection with the controller. These guarantees can be achieved when the identified model provides uncertainty bounds (Hillebrecht & Unger, 2022) or its structure is supported by controller design methodologies (Suykens et al., 1995).

Existing benchmarks either deal with synthetic data only (Bhamidipaty et al., 2023), provide only a single evaluation metric (Schoukens & Noël, 2017; Champneys et al., 2024), or are specialized for a specific model class (Zhong et al., 2021) or dataset (Dulny et al., 2023).

Individual studies introduce domain-specific evaluation criteria, such as the *custom vessel distance measure* (Mathioudakis et al., 2025), *dynamic error budgeting* (Jabben, 2007), or *faithfulness* (Baier et al., 2023). These metrics are designed by domain experts and tailored for a specific system class or require a certain model structure. To the best of our knowledge, no benchmark currently exists that

systematically evaluates identified models based on the requirements developer have on the resulting model. With SYSIDBENCH we rethink the construction of a benchmark for system identification methods by first mapping out a set of requirements. The satisfaction of these requirements should then be used as the selection criterion for an identification method rather than relying on individual metrics. To achieve this novel benchmark, we make the following contributions:

- Extracts domain-independent developer requirements from the system identification literature.
- Integrate dynamic-aware preprocessing by identifying a simple linear model to detect transients, derive a truncation length for efficient parameter optimization, and separating a out-of-distribution (OOD) dataset.
- Systematically evaluates prediction as well as generalization capabilities and tests the identified model on robustness to noisy inputs.

With SYSIDBENCH we provide a benchmark that allows a developer to make an informed decision about which identification algorithm to use.

## 2  RELATED WORK

For an extensive list of identification methods, we refer the reader to the books by Schoukens et al. (2016); Ljung (1998); Pillonetto et al. (2022) and references therein. In these established methods, generalization evaluation is unnecessary because of a bounded model class that is assumed to contain the unknown system. It is the developer's task to hand-pick the model class (Schoukens & Ljung, 2019). When higher-dimensional models are considered for system identification the developer can leave the task of finding a suitable model to the optimizer involved. A prominent example of high dimensional models is neural network archtiectures (Pillonetto et al., 2025) such as recurrent neural networks (Hochreiter & Schmidhuber, 1997; Goodfellow et al., 2016). These models can achieve high prediction accuracy when used for system identification (Mohajerin & Waslander, 2019; Gu et al., 2021; Hu et al., 2024). The high dimensionality comes at the price of overfitting to the training data and the requirements of evaluating generalization capabilities.

The benchmark (Bhamidipaty et al., 2023) consists of 20 synthetic datasets with in-distribution (ID) and OOD evaluation. The nonlinear benchmark[1] (Schoukens & Noël, 2017) is a collection of 13 datasets taken from real physical systems. This benchmark includes system identification baselines both from the systems and control as well as the machine learning community,

The robustness of neural networks, which is considered to be a safety feature, is analyzed in a series of paper that enforce rigorous input-output (Fazlyab et al., 2019; Revay et al., 2020; Pauli et al., 2021) or input-to-state (Bonassi et al., 2021) guarantees. From these methods, we draw inspiration to design empirical robustness measures that are relevant for practical use-cases.

## 3  DEVELOPER REQUIREMENTS

We have synthesized developer requirements for system identification in different application domains by surveying evaluation techniques over an extensive range of system identification papers.

Papers on system identification have commonly measured accuracy by comparing the predictions made by the parametrized model with the outputs given in the dataset. The prediction as well as the measured output signals are in the time domain. (Ljung, 1998; Pillonetto et al., 2022; Forgione & Piga, 2021a; Beintema et al., 2023b; Baier et al., 2023). As a result, we state accurate predictions as the first developer requirement.

**Requirement 1** (Accuracy). *Judging prediction capabilities in the time domain.*

Related work has demonstrated that the system identification methods may fail to predict the correct phase of a function, but predict a phase-shifted function (cf. (Billings, 2013; Pintelon & Schoukens, 2012)). When evaluating the prediction capabilities with root mean squared error (RMSE), Gaussian

---

[1] https://www.nonlinearbenchmark.org/benchmarks

noise seemingly was a better predictor than a phase-shifted ground truth. Therefore, we posit that prediction capabilities must not only be judged by RMSE in the time domain, but also by an evaluation in the frequency domain and state it in the following requirement:

**Requirement 2** (Frequency recovery). *Judging prediction capabilities in the frequency domain.*

Dynamic system models operate in various domains, including acceleration prediction on aircraft wings (Noël & Schoukens, 2020a), displacement estimation of a mass (Frank, 2025), and velocity and angular-velocity prediction for ships (Baier & Staab, 2022). In multi-output models, differences in physical units can strongly influence performance assessment. For instance, a model may appear highly accurate if it predicts forward velocity well, even when its estimates of angular velocity deviate substantially — simply because the two outputs lie on very different scales. We refer to output signals with different physical units as heterogeneous. Consequently, evaluation metrics for system identification models must handle heterogeneous outputs in a way that is independent of their numerical ranges. We capture this need in the following requirement:

**Requirement 3** (Heterogeneity). *Judging prediction capabilities for heterogeneous outputs.*

Simulation models are used to predict the behavior of dynamic systems in unknown scenarios (Lazar, 2024). A model with high generalization capabilities makes accurate predictions when faced with input signals (and initial states) not seen before (Baier et al., 2023). Unknown input signals, used to assess generalization capabilities empirically, are not available in datasets in general. Generalization capabilities are significant for simulation models (Revay et al., 2020; Srivastava et al., 2014; Hu et al., 2024). Since generating new experiments to obtain real OOD data is impractical, we extract a OOD dataset from the existing recordings, allowing us to evaluate generalization capabilities.

**Requirement 4** (OOD Generalization). *Judging generalization capabilities to OOD input signals.*

When identifying dynamic systems in a high-dimensional space, there exists a significant risk of obtaining unstable models, which can lead to unsafe behavior or model failure during inference (Bonassi et al., 2021; Revay et al., 2020). Instability can arise from ID input signals and thus might not be detected by evaluations on the OOD dataset. Therefore, we develop analyses that generate specific input signals to test the robustness of the model with respect to input disturbances. Robustness is the capability of a model to make accurate predictions in the presence of perturbed noisy signals (cf. (Fazlyab et al., 2019; Madry et al., 2017; Pauli et al., 2021; Cohen et al., 2019))

**Requirement 5** (Robustness). *Judging model robustness for disturbed input signals.*

In summary, we found that existing performance comparisons of models for dynamic systems are not suitable for high-dimensional models that outperform classic methods in terms of their predictive accuracy.

## 4  OVERVIEW OF SYSIDBENCH AND NOTATION

We have developed SYSIDBENCH to allow the developer to compare different system identification algorithms according to the judgement criteria stated in the five requirements 1 - 5. An overview of our novel methodology realized in SYSIDBENCH is provided in Figure 1.

SYSIDBENCH uses four publicly available datasets and incorporates six evaluation metrics on five different identification algorithms. Our methodology takes as input signal a dataset of input-output measurements from an unknown dynamic system and a system identification method, and outputs evaluation metrics along with a parametrized model. We assume that the system identification method is a black-box that learns by adapting its model parameters based on the training data. The development of new algorithms is not part of this benchmark.

We denote the raw recording provided by the user as a set of tuples $\mathcal{D} = \{(\boldsymbol{u}, \boldsymbol{y})_i\}_{i=1}^{M}$ that are the input-output measurements taken from the unknown system. Subsets are indicated by a subscript. In particular, we will have two test sets denoted by $\mathcal{D}_{\text{test}} = \mathcal{D}_{\text{ID}} \cup \mathcal{D}_{\text{OOD}}$ for ID and OOD test sets, respectively. While the ID dataset has the same data distribution as the training dataset, the OOD datasets stems from a different distribution. The training and validation sets are denoted by $\mathcal{D}_{\text{train}}$ and $\mathcal{D}_{\text{val}}$. We assume the measurements to be finite and assume trajectories in $\mathcal{D}$ have length $N$, the recordings are not required to have the same length.

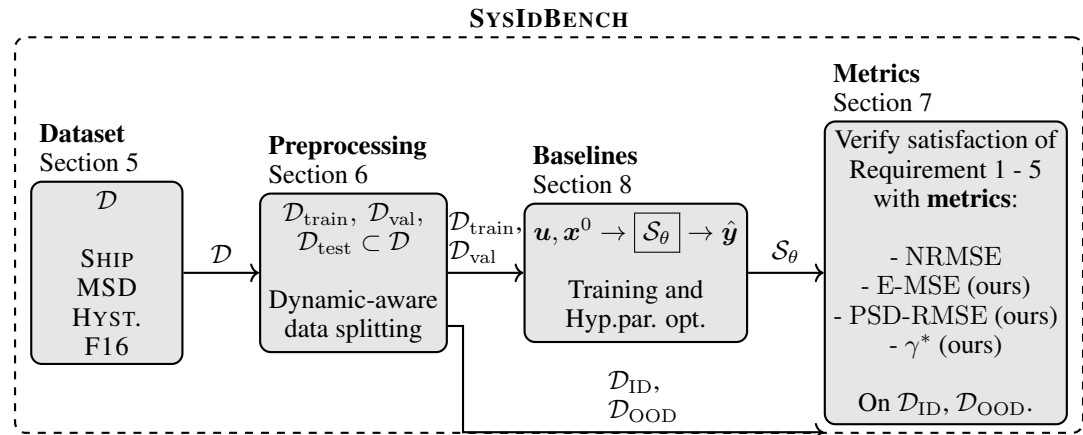

Figure 1: SYSIDBENCH supports systematic evaluation and comparison of system identification methods by assessing the methods according to requirements 1–5. The training dataset $\mathcal{D}_{\text{train}}$ is used to learn the parameters $\theta$, and the validation dataset $\mathcal{D}_{\text{val}}$ is used to optimize the hyperparameters. Test data $\mathcal{D}_{\text{ID}}$, $\mathcal{D}_{\text{OOD}}$, is used for final evaluation.

The algorithm to parametrize a dynamic system model is called an identification or learning algorithm. While identification is rather used in systems and control, learning is the term used in machine learning. We denote a parametrized model by $\mathcal{S}_\theta$ where $\theta \in \mathbb{R}^n$ is a set of parameters to be identified. The model $\mathcal{S}_\theta$ maps an input signal $\boldsymbol{u}$ and an initial state $\boldsymbol{x}^0$ to an output signal $\hat{\boldsymbol{y}} = \mathcal{S}_\theta(\boldsymbol{u}, \boldsymbol{x}^0)$.

The structure of this paper will follow the flow shown in Figure 1, in Section 5 we will given an overview of the considered datasets. We introduce our novel dynamic-aware preprocessing method in Section 6 and discuss the derived metrics for evaluation in Section 7. Baseline results are provided in Section 8.2 and we conclude our paper in Section 9.

## 5 DATA SETS

The four currently available datasets are a ship that moves in open water SHIP, a coupled-mass-spring-damper system MSD, a hysteretic system HYST. and a vibration test of an aircraft wing F16.

The SHIP dataset simulates the normal operation of a ship in open water and provides multiple input and output signals. It is generated from random maneuvers an operator would do. The measured output signals are the velocities in two directions and two rotation rates. The input signals are the propeller speed, the rudder angle, and wind measurements. The complexity of the dynamics arise from a nonlinear mapping between wind and the rotations rates (see Figure 2a). MSD represents a mass-spring-damper system with four masses in which the nonlinearity is static and stems from the force profile of the spring, it has no memory, and has one input and one output (see Figure 2b). The HYST. dataset serves as a toy example that exhibits complex nonlinear behavior, with known underlying differential equations. It has one input and one output (see Figure 2c. F16 consists of dedicated experiments conducted in a controlled environment on a real-world aircraft wing, with one input and multiple output signals. The input signals are carefully designed by experts to excite specific frequencies.

## 6 DATA PREPROCESSING

We differentiate two different kind of datasets, (i) A dataset that provides a split into *training*, *validation*, and *testing*. (ii) A dataset that provides a split into *training*, *validation*, *ID testing*, and *OOD testing*. The type of dataset determines which preprocessing steps are required. In the first step, we determine the sequence length used to identify the dynamic system model.

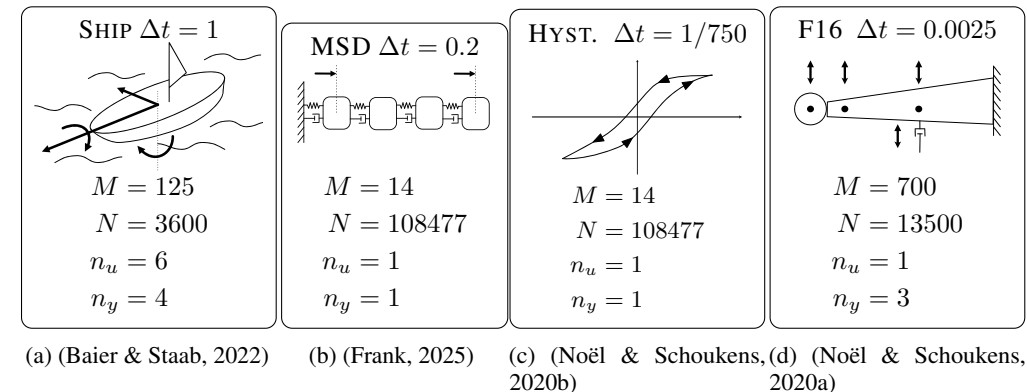

(a) (Baier & Staab, 2022)    (b) (Frank, 2025)    (c) (Noël & Schoukens, 2020b)    (d) (Noël & Schoukens, 2020a)

Figure 2: Overview of datasets used in the benchmark. $M$ refers to the number of recordings that are available, $n_{\boldsymbol{u}}$, $n_{\boldsymbol{y}}$ refer to the number of inputs and output signals, respectively, and $N$ refers to the length of one recording. With $\Delta t$, we refer to the discrete sampling time.

## 6.1 TRUNCATION LENGTH

Optimization on long sequences is computationally expensive and can not be parallelized (Ribeiro et al., 2020; Forgione & Piga, 2020; Beintema et al., 2023a). For models with a large number of parameters, this becomes a bottleneck. In Ribeiro et al. (2020), it was shown that the optimization landscape is smoother on shorter sequences. In Beintema et al. (2023b;a), this was termed the truncation length, describing the length of subsequences used for training. The truncation length is a design parameter that depends on the system's dynamics and the sample time used for discretization. In machine learning related publications, this is often termed the prediction horizon (Baier et al., 2023; Mohajerin & Waslander, 2019).

For our benchmark, we aim for a unifying approach to obtain the truncation length without having prior knowledge about the system. In this dynamic-aware preprocessing, we describe a method to get the truncation length based on the transient time of a linear approximation model. With this approach, we remain general with respect to the system's domain but take into account the dynamics and the sampling time.

After obtaining a linear state space model from the available training data, we use the linear matrices to get the steady state response. Then we apply a step function to the linear model and analyse the output. The time until the linear dynamics reach a steady state is called the transient time. For a linear system, it tells us how long you have to observe the output signal in order to recover its dynamics.

We use N4SID, a subspace identification method (Van Overschee & De Moor, 1994; Ljung, 1998), to obtain a linear state space model of the form:

$$\begin{pmatrix} \boldsymbol{x}_{\text{lin}}^{k+1} \\ \hat{\boldsymbol{y}}_{\text{lin}}^{k} \end{pmatrix} = \begin{pmatrix} \boldsymbol{A}_{\text{lin}} & \boldsymbol{B}_{\text{lin}} \\ \boldsymbol{C}_{\text{lin}} & \boldsymbol{D}_{\text{lin}} \end{pmatrix} \begin{pmatrix} \boldsymbol{x}_{\text{lin}}^{k} \\ \boldsymbol{u}_{\text{lin}}^{k} \end{pmatrix}. \tag{1}$$

Where $\boldsymbol{x}_{\text{lin}}^{k} \in \mathbb{R}^{n_{x_{\text{lin}}}}$ represents the internal state of the linear model, and the input and output sizes match the size of the physical system $n_e = n_{e_{\text{lin}}}$ and $n_d = n_{d_{\text{lin}}}$. From the linear approximation model, we extract the transient time, which is defined as the time it takes for a system to reach a steady state after a step input. The steady state can be computed by solving the eigenvalue problem of the matrix $\boldsymbol{A}_{\text{lin}}$ in equation 1. More details about N4SID can be found in the appendix A.1.

**Definition 1.** *We define the transient time $\tau > 0$ as the time it takes until the linear model's output reaches a steady-state when fed with a step input.*

$$k^* = \min_k \|\hat{\boldsymbol{y}}_{\text{lin}}^k - \bar{\boldsymbol{y}}\| < \epsilon \boldsymbol{y}_{\max} \tag{2}$$

*where $\boldsymbol{y}_{\text{ss}}$ refers to the steady state and*

$$\boldsymbol{y}_{\max} = \max_k \|\hat{\boldsymbol{y}}_{\text{lin}}^k - \bar{\boldsymbol{y}}\| \tag{3}$$

*is the maximum deviation from the steady state. The transient time follows as the minimal time index $k^*$ and the sample time $\Delta t$, $\tau = k^* \Delta t$.*

The transient time is a property of the step response and depends on the dynamics of the model. We use the transient time to obtain the truncation length $h$ used during training. We also derive the window length $w$ used for initialization based on the horizon. We choose $\epsilon = 0.02$, corresponding to reaching the steady state within $2\%$ of the maximum deviation.

More details about the transient-time, how the steady-state is calculated, and the step response, especially for models with multiple inputs and outputs, are found in the appendix A.2.

Based on the truncation length, we can split the raw recordings given in $\mathcal{D}$ into subsequences of length $h + w + 1$ and use them for training. In the next step, we develop a methodology to extract a OOD dataset from the subsequences.

## 6.2 OUT OF DISTRIBUTION DATASET

After the truncation length is fixed, we separate an OOD dataset to assess models' generalization capabilities. The OOD separation is based on the signal energy which is defined as follows:

$$E_u := \|\boldsymbol{u}\|_2^2 = \sum_{k=0}^{N-1} (\boldsymbol{u}^k)^{\mathrm{T}} \boldsymbol{u}^k \text{ for } N > 0. \tag{4}$$

We assume the sequences to be square summable, i.e. $\|\boldsymbol{u}\|_2 < \infty$.

It measures the energy injected into the system by the input signal. The energy is computed for each subsequence of the recordings, and the subsequences with the highest energy are selected for the OOD dataset. This approach allows us to evaluate the model's generalization capabilities without requiring expensive new experiments or expert knowledge about the system.

**Remark 1.** *Conservation of energy is a fundamental property of physical systems. A dynamic system is subject to these properties and converts energy. It is irrelevant which physical states are being considered or influenced. By considering energy, systems from different domains can be compared, making this perspective appropriate for our benchmark.*

The subsequences are selected such that

$$\mathcal{D}_{\mathrm{OOD}} = \{(\boldsymbol{u}^k, \boldsymbol{e}^k) \in \{\mathcal{D}_{\mathrm{test}}, \mathcal{D}_{\mathrm{train}}, \mathcal{D}_{\mathrm{val}}\} \mid \|\boldsymbol{u}\|_2 > \tau_{\mathrm{ood}}\}, \tag{5}$$

The threshold $\tau_{\mathrm{ood}}$ defines the minimum energy required for a subsequence to be included in the OOD dataset. We choose $\tau_{\mathrm{ood}}$ such that the size of the OOD dataset is $10\%$ of $\mathcal{D}$. The remaining subsequences are randomly split into training, validation, and ID testing subsets.

After the dynamic-aware data preprocessing, we get a training dataset $\mathcal{D}_{\mathrm{train}}$ that contains subsequences of length $h + w + 1$, a dataset $\mathcal{D}_{\mathrm{val}}$ for hyperparameter optimization, and two test sets for the final evaluation $\mathcal{D}_{\mathrm{ID}}$ and $\mathcal{D}_{\mathrm{OOD}}$.

# 7 METRICS FOR EVALUATING DEVELOPER REQUIREMENTS

Based on the developer requirements defined in Section 3, we now derive evaluation metrics that allow a developer to verify the requirements are satisfied.

## 7.1 PREDICTION CAPABILITIES IN TIME DOMAIN (REQUIREMENT 1)

To evaluate the accuracy of model predictions in the time domain, we use the normalized root mean squared error (NRMSE) and the fit-metric, which are also used in existing benchmarks (Champneys et al., 2024). The NRMSE measures the deviation per channel.

$$\mathrm{NRMSE}_m(\boldsymbol{u}, \boldsymbol{y}) := \frac{1}{\sigma_m} \sqrt{\frac{1}{Mh} \sum_{i=1}^{M} \sum_{k=0}^{h-1} ((y_m^k)_i - (\hat{y}_m^k)_i)^2} \text{ for } m = 1, \ldots, n_y, (\boldsymbol{u}, \boldsymbol{y}) \in \mathcal{D}_{\mathrm{test}},$$

$$\tag{6}$$

where $\sigma_m = \sqrt{\sum_{i=1}^{M} \sum_{k=0}^{N-1} (((y_m^k)_i - \bar{y}_m))}$ refers to the standard deviation of the output channel and is calculated from the training subsequences. The NRMSE is computed for each output channel $m$ separately, allowing for a more detailed analysis of the prediction quality across different outputs. A low NRMSE *value refers to more accurate predictions.*

## 7.2 Prediction capabilities in the frequency domain (Requirement 2)

While the NRMSE provides a good indication of the quality in the time domain, it might not capture whether the model has learned to reproduce the oscillatory behavior of the system. A model that predicts the mean value of an oscillatory output can achieve a low NRMSE, but it fails to capture the oscillatory nature of the system. To address this limitation, we propose a frequency-based evaluation metric that compares the power spectral density (PSD).

The power PSD of the output signal $\boldsymbol{y}$ is estimated using the Welch (Welch, 1967) method as

$$S_{yy}(f) = \lim_{k \to \infty} \frac{1}{N} \left| \mathcal{F}\{\boldsymbol{y}^k\} \right|^2, \tag{7}$$

where $\mathcal{F}\{\cdot\}$ denotes the discrete Fourier transform (DFT), $k$ is the discrete-time index, $N$ is the number of samples, and $f$ the frequency. The signal $\boldsymbol{y}^k$ is considered in the limit as $N \to \infty$ and the sampling interval approaches zero. Welch's approach partitions the signal into overlapping segments, applies a window function to each, computes the periodogram for each segment, and then averages them to reduce variance in the PSD estimate. The resulting estimate provides a practical measure for frequency-domain analysis and recovery.

To evaluate the difference between the PSD of the true output $\boldsymbol{y}$ and the predicted output $\hat{\boldsymbol{y}}$, we define the RMSE on the PSD as

$$\text{PSD-RMSE} = \sqrt{\frac{1}{N_f} \sum_{i=1}^{N_f} (S_{yy}(f_i) - S_{\hat{y}\hat{y}}(f_i))^2}, \tag{8}$$

where $S_{\hat{y}\hat{y}}(f)$ is the PSD of the predicted output $\hat{\boldsymbol{y}}$, $N_f$ is the number of frequency bins, and $f_i$ represents the $i$-th frequency bin.

## 7.3 Capabilities for heterogenious predictions (Requirement 3)

In addition to the metrics in the time domain equation 6 we propose an alternative metric that captures the deviation in energy and is independent of the physical states of the system. The energy of the output signal is defined in equation 4. The energy error is defined as

$$\text{E-MSE} = \frac{|E_y - \hat{E}_y|}{E_y}, \tag{9}$$

where $E_y$ is the energy of the output signal and $\hat{E}_y$ is the energy of the predicted trajectory.

## 7.4 Generalization to out-of-distribution input signals (Requirement 4)

To evaluate the generalization capabilities of the identified model, we use the OOD test dataset that was separated during preprocessing of the data (see Section 6 for details). We evaluate the OOD dataset on the same metrics as the ID test dataset.

## 7.5 Robustness to disturbed input signals (Requirement 5)

To assess the robustness of a model to unknown input signals, we aim to produce outputs that violate physical laws, such as energy conservation. The setup is comparable to adversarial attacks in classification problems, where the goal is to find a small input perturbation that leads to large changes in the output. In regression problems this is more challenging since there is no true output class. However, we use the energy perspective to design adversarial attacks that lead to output prediction by the model that violate energy conservation.

The first metric to assess model robustness is inspired by the maximum amplification. We aim to find an input signal that maximizes the amplification of energy, i.e., we solve the optimization problem

$$\gamma^* = \max_{\boldsymbol{u}} \frac{\|\hat{\boldsymbol{y}}\|_2}{\|\boldsymbol{u}\|_2} \quad \text{s.t.} \quad \hat{\boldsymbol{y}} = \mathcal{S}_\theta(\boldsymbol{u}, \boldsymbol{x}^0), \quad \|\boldsymbol{u}\|_2 \leq E_{\max} \tag{10}$$

The optimization problem is solved using gradient-based optimization, where the gradient of the output energy with respect to the input signal is computed using backpropagation through the model $\mathcal{S}_\theta$.

In the second metric, we consider the sensitivity as the objective of the opimization. The sensitivity measures how small input changes effect the output of a model. This is also known as the Lipschitz constant

$$L = \max_{(\boldsymbol{u})_a, (\boldsymbol{u})_b} \frac{\|(\hat{\boldsymbol{y}})_a - (\hat{\boldsymbol{y}})_b\|_2}{\|(\boldsymbol{u})_a - (\boldsymbol{u})_b\|_2} \quad \text{s.t.} \quad a \neq b, \ \hat{\boldsymbol{y}} = \mathcal{S}_\theta(\boldsymbol{u}, \boldsymbol{x}^0) \tag{11}$$

# 8 VALIDATION OF THE BENCHMARK WITH SELECTED SYSTEM IDENTIFICATION METHODS

In this section we first present the chosen baslines and then discuss the results when the identification algorithms are used to identify the datasets from Section 5.

## 8.1 SELECTED BASELINE SYSTEM IDENTIFICATION METHODS

Recurrent neural network (RNN) models are infinitely-dimensional (Pillonetto et al., 2025) and thus capable of identifying complex dynamics. We use LSTM and RNN to represent the recurrent neural network architectures, which also cover a wide range of other state space neural network structures, such as (Beintema et al., 2023a; Forgione & Piga, 2020; Mohajerin & Waslander, 2019). The recurrent structure makes the model class flexible, but comes at the price of high computational complexity when dealing with long signals (Beintema et al., 2023a; Ribeiro et al., 2020). Structured state space models (Gu et al., 2021; Gu & Dao, 2023; Hu et al., 2024; Bonassi et al., 2024) overcome this computational bottleneck by simplifying the network structure. These novel models are represented in our benchmark by DEEPSUBENC. Another problem with RNNs is that they overfit the training data. Regularization promises to overcome overfitting by constraining the trainable parameters (Fazlyab et al., 2019; Bonassi et al., 2021; Baier et al., 2023) we represent this model class by RELINET (Baier et al., 2023).

## 8.2 BENCHMARK RESULTS FOR SELECTED BASELINE METHODS

We run the baselines from Section 8.1 on the set of datasets described in Section 5. The results are shown in Table 1 and we report the NRMSE (cf. 6) to assess the accuracy in the time domain, the PSD-RMSE (cf. 8) to assess the frequency recovery, and the E-MSE (cf. 9) to assess the performance on heterogeneous outputs. For judging the robustness, we report the worst-case amplification (cf. 10).

By taking into account multiple evaluation metrics, developers can trade off prediction accuracy versus sensitivity. This is, for example, shown on the MSD dataset, where the RNN and the LSTM have comparable prediction accuracy in terms of NRMSE, but the RNN is significantly more robust in terms of its Lipschitz constant, which is shown by a lower L value.

# 9 CONCLUSION

With SYSIDBENCH, we provide an initial step toward systematic benchmarking system identification methods, enabling developers to select the most suitable approach for their specific problem. We mapped out five key requirements, derived from both the literature and our own experiments.

Our experiments show that high-dimensional models are capable of capturing complex dynamics and achieving high prediction accuracy in both the time and frequency domains. This aligns with findings from the system identification literature. However, these models also exhibit a downside: they can become highly sensitive, as indicated by the metrics for Requirement 5.

Overall, SYSIDBENCH lays the groundwork for systematic evaluation of dynamic system models, while offering clear avenues for future extension and refinement.

Table 1: Comparison of different identification methods across different datasets

| Method | Dataset | NRMSE 6 ID | NRMSE 6 OOD | PSD-RMSE 8 ID | PSD-RMSE 8 OOD | E-MSE 9 ID | E-MSE 9 OOD | $\gamma^*$ 10 | L 11 | #Pars |
|---|---|---|---|---|---|---|---|---|---|---|
| SHIP | LSTM | [0.50 0.43 1.20 0.34] | [0.67 0.56 0.86 0.30] | 0.50 | 1.12 | 0.14 | 0.04 | 65.39 | 154.89 | 334340 |
|  | RNN | [0.41 0.35 1.29 0.31] | [0.50 0.55 0.96 0.30] | 0.31 | 0.84 | 0.08 | 0.03 | 41.15 | 13.85 | 83972 |
|  | S4 | [0.10 0.20 1.00 0.20] | [0.33 0.49 0.99 0.22] | 0.18 | 0.67 | 0.00 | 0.02 | 11.77 | 8.88 | 9338 |
|  | ReLiNet | - | - | - | - | - | - | - | | |
| MSD | LSTM | [0.18] | [0.40] | 6.09 | 61.59 | 0.06 | 0.23 | 2.06 | 46.58 | 331393 |
|  | RNN | [0.21] | [0.30] | 6.28 | 54.41 | 0.11 | 0.18 | 3.49 | 10.27 | 82945 |
|  | S4 | [0.26] | [1.05] | 12.15 | 134.87 | 0.06 | 0.42 | 1.24 | 130.97 | 2887 |
|  | ReLiNet | - | - | - | - | - | - | - | | |
| HYST. | LSTM | [0.38] | [0.39] | 0.00 | 0.00 | 0.01 | 0.03 | 11.60 | 23.48 | 331393 |
|  | RNN | [0.38] | [0.39] | 0.00 | 0.00 | 0.02 | 0.04 | 4.64 | 29.31 | 82945 |
|  | S4 | [0.47] | [0.45] | 0.00 | 0.00 | 0.03 | 0.16 | 3.07 | 36.35 | 2887 |
|  | ReLiNet | [0.42] | [0.40] | 0.00 | 0.00 | 0.15 | 0.13 | 7.28 | 4.04 | 420736 |
| F16 | LSTM | [0.26 0.35 0.40] | [0.36 0.50 0.59] | 3.25 | 11.73 | 0.02 | 0.16 | 5.90 | 8.32 | 331651 |
|  | RNN | [0.41 0.60 0.65] | [0.43 0.72 0.79] | 6.09 | 17.13 | 0.02 | 0.04 | 29.64 | 1284.44 | 83203 |
|  | S4 | [0.57 0.81 0.84] | [0.54 0.81 0.86] | 11.38 | 20.95 | 0.58 | 0.28 | 2.73 | 2.59 | 2891 |
|  | ReLiNet | - | - | - | - | - | - | - | | |

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

## A APPENDIX

### A.1 N4SID ALGORITHM

The Numerical Subspace State Space System Identification (N4SID) algorithm is a subspace-based method for identifying state-space models directly from input–output data. It constructs block Hankel matrices from measured signals, employs numerical linear algebra techniques such as singular value decomposition (SVD) to estimate the system order, and computes state sequences in a least-squares sense. From these estimates, the system matrices of a minimal realization are obtained without requiring nonlinear optimization. N4SID is valued for its numerical robustness, computational efficiency, and ability to handle multivariable systems.

Table 2: Experiment configuration

| Dataset | Model | $n_h$ | Epochs | Batch Size | Learning Rate | loss | $n_u$ | $n_y$ | dropout | # layers |
|---------|-------|-------|--------|------------|---------------|------|-------|-------|---------|----------|
| F16 | lstm | 128 | 1000 | 16 | 0.001 | mse | 1 | 3 | 0.25 | 3 |
| F16 | rnn | 128 | 1000 | 16 | 0.001 | mse | 1 | 3 | NaN | 3 |
| F16 | s4 | 128 | 1000 | 16 | 0.001 | mse | 1 | 3 | NaN | 5 |
| hyst | lstm | 128 | 1000 | 16 | 0.001 | mse | 1 | 1 | 0.25 | 3 |
| hyst | rnn | 128 | 1000 | 16 | 0.001 | mse | 1 | 1 | NaN | 3 |
| hyst | s4 | 128 | 1000 | 16 | 0.001 | mse | 1 | 1 | NaN | 5 |
| msd | lstm | 128 | 1000 | 16 | 0.001 | mse | 1 | 1 | 0.25 | 3 |
| msd | rnn | 128 | 1000 | 16 | 0.001 | mse | 1 | 1 | NaN | 3 |
| msd | s4 | 128 | 1000 | 16 | 0.001 | mse | 1 | 1 | NaN | 5 |
| ship | lstm | 128 | 1000 | 16 | 0.001 | mse | 6 | 4 | 0.25 | 3 |
| ship | rnn | 128 | 1000 | 16 | 0.001 | mse | 6 | 4 | NaN | 3 |
| ship | s4 | 128 | 1000 | 16 | 0.001 | mse | 6 | 4 | NaN | 5 |

## A.2 TRANSIENT TIME

We are interested in the case $x_{\text{lin}}^{k+1} = x_{\text{lin}}^k$ and the respective output. Solving the linear equation leads to the steady-state output

$$\bar{e}_{\text{lin}} = \left( D_{\text{lin}} - C_{\text{lin}} \left( A_{\text{lin}} - I \right)^{-1} B_{\text{lin}} \right) d \tag{12}$$

## A.3 EXAMPLES OF ENERGY RESPONSES

## A.4 ENERGY-TO-PEAK GAIN

The energy-to-peak gain of a linear, stable system characterizes the maximum instantaneous output amplitude in response to a finite-energy input. In the stochastic setting, it describes how large the system's response can become when excited by white noise. The energy-to-peak gain focuses specifically on the largest possible transient. In system identification the energy-to-peak gain is valuable for assessing physical plausibility. For instance, a real ship, due to its mass and damping, responds gradually to small random rudder inputs. A learned model with a high energy-to-peak gain might instead produce sharp, nonphysical yaw deviations—indicating that, despite low average error, it fails to reproduce key dynamic behaviors.

## A.5 EXPERIMENTAL SETUP

The hyperparameters used for our experiments are shown in Table 2

## A.6 POWER SPECTRAL DENSITY EXAMPLE

In this section, we compare the predictions of a trained model with those of a noise baseline. In the time domain, the RMSE shows that predicting noise performs only slightly worse than the trained model. In the plots of Figure 4 one observes a slight phase shift made by the LSTM which leads to errors that are comparable to predicting noise, even though the correct frequency is recovered. This observation motivates a deeper analysis in the frequency domain. Table 3 reports the mean error across all outputs in both domains. While the errors in the time domain are comparable, the LSTM clearly outperforms the noise baseline in the frequency domain. This advantage is also evident in Figure 3, where the LSTM accurately recovers frequencies up to 15 Hz, whereas the noise model fails to capture the true frequency components.

Table 3: Comparison between prediction made by an `LSTM` and a noise model in time and frequency domain on the F16-GVT example (Noël & Schoukens, 2020a).

| Model | RMSE | PSD-RMSE |
|-------|------|----------|
| Noise | 1.22 | 36.70 |
| LSTM | 1.17 | 24.92 |

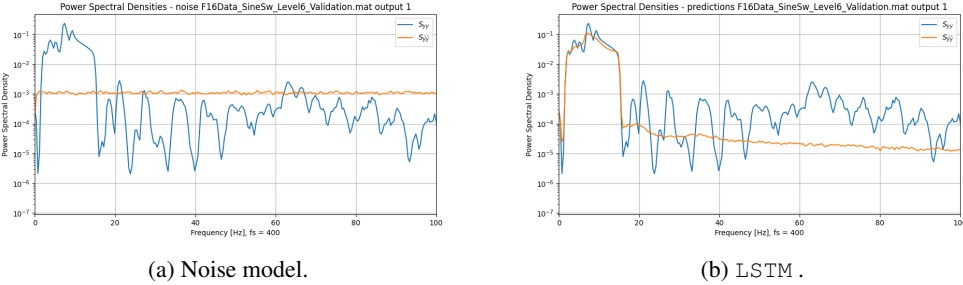

(a) Noise model.    (b) `LSTM` .

Figure 3: Power spectral density over frequencies.

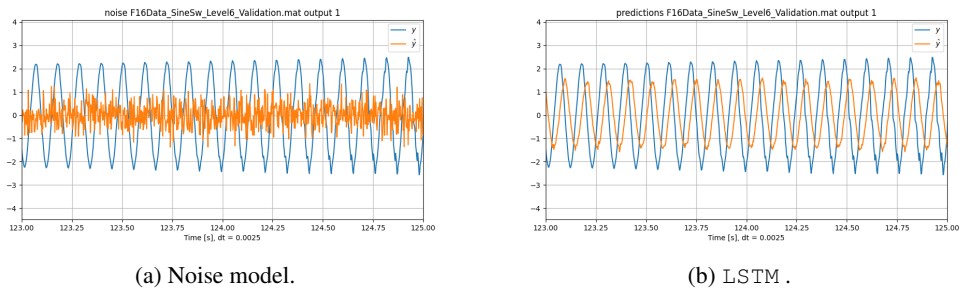

(a) Noise model.    (b) `LSTM` .

Figure 4: Prediction in time domain.