# OpenReview forum: "SysIdBench: A Benchmark for System Identification Methods"
_ICLR.cc/2026/Conference — ICLR 2026 Conference Withdrawn Submission_

### Official Review · Reviewer_TGvv · 2025-10-26

**Soundness:** 2
**Presentation:** 1
**Contribution:** 1
**Rating:** 2
**Confidence:** 4

**Summary:**

The paper proposes a new benchmark for system identification. The authors rigorously identify requirements for benchmark evaluation. A small set of data sets is provided and empirically evaluated on a small set of methods.

**Strengths:**

- the idea of unifying a system identification benchmark in the era of machine learning method is sound
- the systematic identification of requirements for metric selection is rigorous

**Weaknesses:**

- The benchmark provides only a small set of four data sets which already exist
- There is no comparison to existing classical system identification methods
- The paper draws no conclusion about the use of metrics, the type of preprocessing, or the fitting method across different settings

**Questions:**

- line 079: "archtiectures" misspelled
- in related work, can you relate the existing benchmarks to your work?
- line 89: "paper" should be plural. There is a number of more spelling mistakes. I urge the authors to rigorously double check the manuscript.
- line 95/96: "We have synthesized developer requirements for system identification in different application domains by surveying evaluation techniques over an extensive range of system identification papers." What is the surveying criterion?
- section 3: can you translate the stated (soft) requirements into precise mathematical formulations?
- line 141: "we found that existing performance comparisons of models for dynamic systems are not suitable for high-dimensional models that outperform classic methods". How did you find that with the analysis of the requirements in section 3?
- line 159: "While the ID dataset has the same data distribution as the training dataset, the OOD datasets stems from a different distribution.". How do you define a different distribution here? Is it on input sequence, output sequence, and what is defines the sequence to be different? Some KL-divergence > X?
- What about non-parametric identification approaches? How are they accounted for in your methodology?
- is the F16 dataset taken from the nonlinear system identification benchmark or are these new datasets? What about the other data sets?
- section 6.1:
  - how do you select the orer of the system?
  - you fit a linear model to nonlinear or potentially switching dynamics systems. How can you ensure that (a) your model is meaningful, (b) the step response that you obtain is somewhat correct?
  - Can you provide empirical results and possibel failure cases of your truncation length algorithm?
- section 6.2: What is the theoretical justification for defining OOD samples based on input sequence energy? One could theortically compe up with examples where a widely different input sequence energy leads to similar output sequences.
- line 323: "valuereferstomoreaccuratepredictions"
- line 397: "Recurrent neural network (RNN) models are infinitely-dimensional". Can you justify this? Or what dimensionality are you defining here? RNNs still have a finite dimensional state and transition matrices.
- Can you define more clearly what ReLiNet is?
- Table 1:
  - why are there 4 values for some dataset on ID and OOD NRMSE values?
  - Why are values for ReLiNet only provided for the Hyst dataset?
  - the column name for method and dataset are switched
  - Can you provide performance of classical system identification method, i.e., non machine learning based method on these datasets? E.g. I am aware that there is a range of papers dealing with the f16 dataset from the nonlinear system identification benchmark.
- is the benchmark publicly available, how are new methods evaluated or ranked, will there be a leaderboard, ...?

---

### Official Review · Reviewer_9Crf · 2025-10-28

**Soundness:** 1
**Presentation:** 2
**Contribution:** 1
**Rating:** 2
**Confidence:** 4

**Summary:**

SYSIDBENCH: A BENCHMARK FOR SYSTEM IDENTIFICATION METHODS

The document presents SYSIDBENCH, a novel benchmark for systematically evaluating system identification methods based on developer requirements and various performance metrics.

SYSIDBENCH is a novel benchmark designed to systematically evaluate and compare various system identification methods based on specific developer requirements. ​

SYSIDBENCH addresses the challenges of selecting suitable system identification methods due to the variety of existing algorithms and system characteristics. ​
It incorporates tailored datasets, data splits, and evaluation metrics to enhance benchmarking capabilities.
The benchmark focuses on five key requirements: accuracy, frequency recovery, heterogeneity, out-of-distribution generalization, and robustness.
The code and datasets for SYSIDBENCH are publicly available for developers.

**Strengths:**

SYSIDBENCH provides a nice summary of several data sets and tries to adopt a systematic approach. It  is built upon five essential requirements derived from the literature on system identification. ​

Requirement 1: Accuracy in time domain predictions is crucial for evaluating model performance. ​
Requirement 2: Frequency recovery ensures that models can reproduce oscillatory behaviors accurately. ​
Requirement 3: Heterogeneity addresses the need for metrics that handle outputs with different physical units. ​
Requirement 4: Out-of-distribution (OOD) generalization evaluates model performance on unseen input signals. ​
Requirement 5: Robustness assesses how well models perform under input disturbances. ​

SYSIDBENCH employs a structured methodology to evaluate system identification algorithms against the defined requirements. ​

It utilizes four publicly available datasets: SHIP, MSD, HYST, and F16, each representing different dynamic systems. ​
The methodology includes dynamic-aware preprocessing to optimize training and validation datasets. ​
Six evaluation metrics are used to assess the performance of five different identification algorithms. ​
The benchmark outputs evaluation metrics alongside the identified model parameters.

**Weaknesses:**

The English is poor and needs attention to bring it up to publication standards. It obscures understanding of the article.

The objectives of this article are quite unclear. The authors claim: "no benchmark currently exists that systematically evaluates identified models based on the requirements developer have on the resulting model." However, sytsem identification can be applied is so many different ways, and the algorithms employed are quite different for the different applications. Typically for design you need highly accuracte model, but for online use (diagnostics) less accuracy is required. Some uses are:

 1. Control System Design

Model-based control: Accurate models are needed for designing controllers like PID, LQR, MPC, etc.
Adaptive control: Online system identification helps update models in real-time for systems with changing dynamics.
Robust control: Identification helps quantify uncertainty and design controllers that can handle model variations.


2. Fault Detection and Diagnostics

Residual generation: Compare measured outputs with model predictions to detect anomalies.
Mode identification: As you mentioned, hybrid systems require identifying discrete modes (e.g., gear shifts, valve states).
Health monitoring: Track system degradation over time using identified parameters.


3. Prediction and Forecasting

Time series modeling: ARX, ARMAX, state-space models for forecasting future behavior.
Energy systems: Predict demand or generation in smart grids.
Economics and finance: Identify models for forecasting market trends or consumer behavior.


4. Simulation and Digital Twins

Virtual prototyping: Use identified models to simulate system behavior before physical implementation.
Digital twins: Real-time models that mirror physical systems for monitoring and optimization.


5. Machine Learning Integration

Feature extraction: Identified parameters can serve as features for classification or regression tasks.
Hybrid modeling: Combine physics-based models with data-driven approaches (e.g., grey-box models).
Model learning: Use system identification as a form of supervised learning where the system dynamics are the target.


6. System Understanding and Exploration

Scientific discovery: Identify governing equations or dynamics from experimental data.
Model validation: Compare different hypotheses about system behavior.
Parameter estimation: Infer physical parameters (e.g., mass, damping) from observed data.

7. Process Optimization

Industrial processes: Identify models to optimize throughput, energy use, or quality.
Batch processes: Model transitions between phases for better scheduling and control.


Given this background, what is the purpose of SYSIDBENCH? Section 3 (DEVELOPER REQUIREMENTS) addresses this but it should be obvious that you cannot have a SINGLE benchmark for every purpose. The 5 requirements are poorly developed, and the level of technical specification of these requirements is weak.

What is the underlying model class? This is never addressed. Is it:

-- a simple dynamical system
-- a hybrid dynamical system
-- a distributed dynamical system

I suspect you need different benchmarks for each. Further for the 6 applications listed above I argue that you need a purpose-specific benchmark.

The Benchmark Results for Selected Methods also show how poorly conceived this paper is. Consdider diagnostics: we want to diagnose faults within some time frame, so how is this captured? For ealth monitoring: we want to track system degradation over time using identified parameters, and accurately estimate remaining useful life. So how is this captured?

**Questions:**

1. How is it possible to cover all possible uses of systems identification with one benchmark?

2. Please address how your benchmark enables accurate diagnostics and prognostics development? Don't you need fault data specifically?

---

### Official Review · Reviewer_vZQs · 2025-10-30

**Soundness:** 2
**Presentation:** 2
**Contribution:** 2
**Rating:** 2
**Confidence:** 4

**Summary:**

The paper presents a new benchmark for system identification methods called SysIdBench. Having a good benchmark in any field is crucial, as it enables the systematic study of existing methods and facilitates the transfer of this knowledge to real-world problems. The benchmark was created based on synthesized developer requirements across various applications of system identification methods, including accuracy, frequency recovery, heterogeneity, out-of-distribution generalization, and robustness. It includes the results of four different system identification methods evaluated on four datasets, using five performance metrics across both in-distribution and out-of-distribution test sets. The main contribution lies in defining five evaluation metrics related to the five developer requirements. However, the paper lacks sufficient detail to justify the design choices made in the experimental setup—for example, the selection of datasets, the choice of benchmarked methods, and the approach used for splitting the data into training, validation, and test sets. These aspects require more careful consideration, and different strategies should be analyzed to assess sensitivity. Furthermore, the performance analysis is not adequately conducted. A deeper investigation is needed to demonstrate the importance of the proposed benchmark, including whether the datasets are correlated based on the newly defined metrics. If the dataset distributions in the meta-space are too similar, the benchmark may yield biased results. The main issue with benchmarking today is that many contributions introduce new benchmarks without a clear understanding of their strengths and weaknesses. These are crucial steps that are missing from the paper. Therefore, I recommend that the paper be rejected.

**Strengths:**

The paper introduces a new benchmark for system identification methods, comprising four methods evaluated on four datasets using five performance metrics. These metrics correspond to key development requirements for system identification across different application domains, making the contribution highly relevant for advancing the field.

The originality of the paper lies in defining five evaluation metrics aligned with the five development requirements for system identification—accuracy, frequency recovery, heterogeneity, out-of-distribution generalization, and robustness. This represents a significant improvement over existing benchmarks, which typically focus on a single metric, most often accuracy.

**Weaknesses:**

The paper lacks sufficient arguments regarding the criteria used to select the four datasets included in the benchmark. Could you please also elaborate on how these datasets differ from those used in other benchmarks for system identification?
The paper also does not provide justification for the choice of the method used for determining the truncation length. It reads as if one method was arbitrarily chosen—what would happen if a different method were used, and how would that influence the benchmark results?
Regarding the out-of-distribution datasets, it is stated that 10% of the signals with higher energy are included in the out-of-distribution set. How would the results change if this percentage were increased or decreased? Additionally, it is mentioned that the remaining signal sequences are randomly split into training, validation, and in-distribution test subsets. How many random splits were performed? What would happen if the split were stratified based on energy distribution? How would that influence the benchmark outcomes?
Could you also provide justification for the selection criteria of the four methods included in the experiments?
It appears that the paper simply runs four system identification methods on four datasets and reports five evaluation metrics. However, the paper lacks any analysis of the benchmark itself—for instance, whether the included datasets are correlated across performance metrics, how they differ in their distributions within the meta-space, or what the signal energy distributions look like across datasets and between training, validation, and test splits. These analyses are crucial to determine whether the proposed benchmark truly contributes to benchmarking practices in the field.

**Questions:**

Please elaborate on the details behind the experimental design choices, such as the selection of datasets, methods, and the different training, validation, and test splits.
Please further analyze the benchmark results to identify and discuss the strengths and weaknesses of the proposed benchmark.

**Details Of Ethics Concerns:**

N/A.

---

### Official Review · Reviewer_thFi · 2025-10-31

**Soundness:** 3
**Presentation:** 2
**Contribution:** 2
**Rating:** 4
**Confidence:** 3

**Summary:**

The paper introduces SYSIDBENCH, a benchmark framework for evaluating system identification methods - algorithms that learn dynamic system models from input–output data. Unlike existing benchmarks that are often domain-specific, rely on synthetic data, or use limited metrics, SYSIDBENCH aims to provide a comprehensive, principle-based, and domain-independent evaluation.

The authors define five developer-oriented requirements for evaluating system identification methods:
1. Accuracy in the time domain
2. Frequency Recovery: Evaluate predictions in the frequency domain
3. Heterogeneity: Ensure fair evaluation across outputs with different physical units
4. OOD Generalization: Evaluate model generalization to unseen input distributions
5. Robustness: Assess model stability and performance under perturbed or noisy inputs

**Strengths:**

1. Novel benchmark design integrating energy-related principles
2. Five requirements are motivated by the literature
3. Energy-based OOD dataset extraction is interpretable
4. The paper structure is clear

**Weaknesses:**

1. Evaluation focuses primarily on neural network models, lacking classical baselines like arx
2. No statistical analysis (variance or significance tests) in the reported results
3. The results could be described with more informative figures
4. Not clear if the OOD and the robustness requirements are conceptually distinct: both feed some unusual inputs into the model
5. Not clear how the energy error is related to the heterogeneous prediction in Sec 7.3

**Questions:**

1. Could you provide some meaningful plots showing the generalization (Requirement 4)?
2. Could you provide meaningful examples showing robustness (Requirement 5)?
3. Could you provide the missing baselines and statistical uncertainty in the baseline performance metrics?

---

### Note · Authors · 2025-11-28

I have read and agree with the venue's withdrawal policy on behalf of myself and my co-authors.